# Development and validation of a prediction model for tocilizumab failure in hospitalized patients with SARS-CoV-2 infection

Cristina Mussini[1,2‡]*, Alessandro Cozzi-Lepri[3‡], Marianna Menozzi[1], Marianna Meschiari[1], Erica Franceschini[1], Jovana Milic[2], Lucio Brugioni[4], Antonello Pietrangelo[5], Massimo Girardis[2,6], Andrea Cossarizza[5], Roberto Tonelli[7], Enrico Clini[5,7], Marco Massari[8], Michele Bartoletti[9], Anna Ferrari[10], Anna Maria Cattelan[10], Paola Zuccalà[11], Miriam Lichtner[11], Roberto Rossotti[12], Enrico Girardi[13], Emanuele Nicastri[13], Massimo Puoti[13,14], Andrea Antinori[13], Pierluigi Viale[9], Giovanni Guaraldi[1,2]

**1** Department of Infectious Diseases, Azienda Ospedaliero-Universitaria, Policlinico of Modena, Modena, Italy, **2** Department of Surgical, Medical, Dental and Morphological Sciences, University of Modena and Reggio Emilia, Modena, Italy, **3** Centre for Clinical Research, Epidemiology, Modelling and Evaluation (CREME), Institute for Global Health, UCL Population Health Sciences, University College London, London, United Kingdom, **4** Internal Medicine Department, Azienda Ospedaliero-Universitaria, Policlinico of Modena, Modena, Italy, **5** Department of Medical and Surgical Sciences for Children and Adults, University of Modena and Reggio Emilia, Modena, Italy, **6** Department of Anaesthesia and Intensive Care Unit, Azienda Ospedaliero-Universitaria, Policlinico of Modena, Modena, Italy, **7** Respiratory Diseases Unit, Azienda Ospedaliero-Universitaria, Policlinico of Modena, Modena, Italy, **8** Infectious Disease Unit, Azienda USL-IRCCS di Reggio Emilia, Reggio Emilia, Italy, **9** Department of Medical and Surgical Sciences, University of Bologna, Bologna, Italy, **10** Infectious Disease Unit, Azienda Ospedale, University of Padua, Padua, Italy, **11** Department of Public Health and Infectious Disease, Sapienza University of Rome, Polo Pontino, Italy, **12** Niguarda Hospital Milan, Milano, Italy, **13** National Institute for Infectious Diseases L. Spallanzani (INMI), Rome, Italy, **14** School of Medicine, Università degli studi di Milano Bicocca, Milano, Italy

‡ These authors share first authorship to this work.
* cristina.mussini@unimore.it

**Data Availability Statement:** All relevant data are within the paper and its Supporting Information files.

## Abstract

### Background

The aim of this secondary analysis of the TESEO cohort is to identify, early in the course of treatment with tocilizumab, factors associated with the risk of progressing to mechanical ventilation and death and develop a risk score to estimate the risk of this outcome according to patients' profile.

### Methods

Patients with COVID-19 severe pneumonia receiving standard of care + tocilizumab who were alive and free from mechanical ventilation at day 6 after treatment initiation were included in this retrospective, multicenter cohort study. Multivariable logistic regression models were built to identify predictors of mechanical ventilation or death by day-28 from treatment initiation and β-coefficients were used to develop a risk score. Secondary outcome was mortality. Patients with the same inclusion criteria as the derivation cohort from 3 independent hospitals were used as validation cohort.

**Funding:** This work was supported by Progetto Covid 202012371675 funded by Italian Ministry of Health.

**Competing interests:** The authors have declared that no competing interests exist.

## Results

266 patients treated with tocilizumab were included. By day 28 of hospital follow-up post treatment initiation, 40 (15%) underwent mechanical ventilation or died [26 (10%)]. At multivariable analysis, sex, day-4 $PaO_2/FiO_2$ ratio, platelets and CRP were independently associated with the risk of developing the study outcomes and were used to generate the proposed risk score. The accuracy of the score in AUC was 0.80 and 0.70 in internal validation and test for the composite endpoint and 0.92 and 0.69 for death, respectively.

## Conclusions

Our score could assist clinicians in identifying, early after tocilizumab administration, patients who are likely to progress to mechanical ventilation or death, so that they could be selected for eventual rescue therapies.

## Introduction

After 10 months of SARS-CoV-2 pandemic, it is still unclear which should be the best standard of care for the treatment of COVID-19. On the basis of large randomized clinical trials, only glucocorticoids were shown to be able to reduce mortality, especially among patients undergoing mechanical ventilation [1]. Nevertheless, mortality rate in the RECOVERY trial remained high at 20%, implying that salvage therapies are needed after the failure of this treatment [1]. A few immunomodulatory agents, mainly tocilizumab, anakinra and baricitinib, have been tested in observational studies on the basis of their possible activity against the cytokine storm characterizing more severe COVID-19 clinical pictures and results have been encouraging [2–5]. Tocilizumab is a recombinant humanized monoclonal antibody, of the IgG1 class, directed against both the soluble and the membrane bound IL-6 receptor [6]. A systematic review and meta-analysis of 10 observational studies including 1358 patients demonstrated that mortality was 12% lower for COVID-19 patients treated with tocilizumab compared to patients who were not treated. The number needed to treat was 11, suggesting that for every 11 COVID-19 patients treated with tocilizumab, 1 death could be prevented [7]. These results were confirmed by two more recently published meta-analyses including a larger number of studies [8, 9].

Tocilizumab was also tested in randomized clinical trials with discordant results. A pharmaceutical company press releases showed no evidence of clinical improvement and reduced risk of mortality when comparing patients treated with tocilizumab vs. placebo in the double-blind COVACTA trial [10], but final results remain unpublished. Similar negative results were obtained prescribing tocilizumab to patients with mild to moderate COVID-19 pneumonia in the attempt to prevent the cytokine storm [11, 12]. In contrast, a 44% reduced risk of mechanical ventilation in people treated with tocilizumab was observed in the EMPACTA trial which included patients with a greater severity of disease at enrollment and a significant enrichment in Hispanic, Native American and Black ethnic minorities as compared to the target population examined in COVACTA [13]. Additionally, CORIMUNO, a French multi-centre randomized clinical trial including 129 patients (65 standard of care + 64 tocilizumab), also released a statement regarding the beneficial effect of tocilizumab vs. standard of care in reducing mechanical ventilation and/or mortality at 14 days [14]. More recently, the REMAP-CAP trial conducted in critically ill patients showed a reduction in mortality in patients receiving

either tocilizumab or sarilumab. As a consequence of these results, tocilizumab is now recommended for use in the clinics for patients in critical conditions in the UK [15].

Discordant results may be explained by different inclusion criteria, study methods, faulty interpretation of trial results when sample size was small (confusing 'no statistical significance' with 'no effect of the drug') and overall clinical competence in managing severe COVID-19 patients. Indeed, even before the publication of the REMAP-CAP results, the drug has been largely used in clinical practice worldwide as salvage treatment for critical pneumonia in patients failing standard of care. Nevertheless, there is still an appreciable percentage of patients who fail to respond to tocilizumab, for example in the TESEO cohort by day 28, 15% of patients treated with tocilizumab underwent mechanical ventilation or death [4].

Therefore, as many more people will be now treated with this drug, it is crucial to early identify patients who are likely to fail tocilizumab, so that treatment could be intensified or modified. The aim of this analysis was to identify early predictors of tocilizumab failure in order to develop and validate a risk score to predict mechanical ventilation or death within 28 days of follow-up.

## Methods

This is a secondary analysis of the Tocilizumab in Patients with Severe COVID-19 Pneumonia (TESEO) Study [4], restricted to people treated with tocilizumab. It is a retrospective, observational cohort study done in three tertiary care hospitals in the Emilia-Romagna region, Italy, on patients with COVID-19 severe pneumonia. All consecutively enrolled adult patients ($\geq$18 years) with severe COVID-19 pneumonia, defined by the presence of at least one of the following: a respiratory rate (RR) $\geq$ 30 breaths per minute (bpm), peripheral blood oxygen saturation (SaO$_2$) $\leq$ 93%, a PaO$_2$/FiO$_2$ ratio < 300 mmHg in room air and lung infiltrates >50% within 24–48 hours were included in the study [16, 17].

All patients with moderate and severe COVID-19 disease requiring hospital admission were considered for analysis. Patients were admitted from 12 March 2020 to 28 July 2020. The retrospective data were fully anonymized (the only sensitive data was year of birth) and were assessed on 9 September 2020. The sample of patients analyzed is representative, as the majority of COVID-19 cases during the first wave of epidemic were observed in Lombardy, Veneto and Emilia-Romagna.

A non-randomly selected subset of patients was selected to receive tocilizumab after failing standard of care (SoC) of hydoxycloroquine, lopinavir and low molecular weight heparin. Tocilizumab was given at the dose of 8 mg/kg given intravenously twice 12 hours apart and subcutaneously at a dose of 162 mg administered in two simultaneous doses. The study was approved by the Regional Ethical Committee of Emilia Romagna. This analysis focuses on the subset of people in TESEO who were treated with SoC+ tocilizumab.

### Statistical analysis

For a subset of markers collected in at least one of the tertiary care hospitals contributing data to this analysis, we defined the following 3 time points: baseline, day4 and day9 values. Baseline was defined as the most recent value collected prior to the initiation of tocilizumab. Day 4 value was the most recent value in the time window [+2;+6] days from baseline and day 9 was defied as the most recent value in the time window [+7;+11] days from tocilizumab infusion. We also calculated the two changes, as compared to baseline levels, at day 4 and day 9, respectively.

The main outcome was initiation of mechanical ventilation or death by day 28 from the date of starting tocilizumab and the secondary outcome was death. Of note, at day 6 from

starting tocilizumab participants had to be alive and free from invasive mechanical ventilation to be included in the analysis. This is because day6 was the upper limit of the defined day4 window. A similar selection was used for the day9 values but not used for the main analysis. People who developed the event over follow-up and up to day 28 from starting the drug, were labelled as cases and the remaining controls.

Proportion of females was described and compared by case-control status. Mean and standard deviation (SD) of age, sequential organ failure assessment (SOFA) score and a set of biomarkers at each of the 3 time points were also compared in cases and controls. Chi-square test was used to compare proportions and unpaired t-test was used to compare mean values.

Factors who were most strongly associated (by chi-square and t-test) with the primary endpoint in univariable analysis were selected as candidates to be included in a prediction score. Kaplan-Meier method was used to estimate the percentage of patients still event-free starting from 4 days after tocilizumab initiation.

Univariable and multivariable associations between these factors and the risk of outcome were estimated by means of logistic regression and the magnitude of the association expressed by means of Odds Ratios (OR).

Participants were grouped according to the median value of the markers and ORs of the primary endpoint associated with a value below/above the median were shown.

In order to evaluate the goodness of fit of the model and its ability to predict the outcomes, the area under the receiver operating characteristic (ROC) curve was calculated. This was first calculated in the training set (including all patients in TESEO) and compared to the classifier with AUC = 0.5 using a Mann-Whitney statistics and chi-square test. In order to control for extra-sample variation, a leave-one-out cross validation (CV) was implemented. This amounts to a K-fold cross validation, with K equal to the total number of participants N. This means that N separate times, the function approximator is trained on all the data except for one point and a prediction is made for that point. The AUC in CV was computed and used to evaluate the predictive ability of the model by comparing it with the value obtained on the training set.

In addition, we used an independent sample of patients seen in other three Italian hospitals with identical inclusion criteria and definition of outcomes (Niguarda-Milano, INMI-Rome and Padua) to test the score on external data (the test set).

Symmetrical analyses were performed for the primary and secondary endpoints.

For the purpose of simplicity and to broadly categorize the risk of failing tocilizumab, the study population was divided into those with low (0–10%), moderate (11–20%), and high (>20%) risk of day-28 mechanical ventilation/death. These groups were matched to actual risk ranges calculated from the propensity score formula below (1). Exact individual's risk can be calculated *as per* this formula by entering the individual's own demographics and biomarkers values.

Prob (tocilizumab failure) = $\theta$ / (1+ $\theta$) where $\theta$ = exp ($\beta_0 + \beta_1 X_1 + \beta_2 X_2$), $X_1$, $X_2$, etc., are the patients' characteristic values and $\beta_0$, $\beta_1$, $\beta_2$, etc., are the parameter estimates from the logistic regression model.

## IRB approval and data information

The study was approved by *Regional ethical committees* of Emilia Romagna (Modena and Bologna), Lombardy (Niguarda-Milano), Lazio (INMI-Rome) and Veneto (Padua).

In Modena and Bologna all tocilizumab-treated patients provided verbal, not written, informed consent due to isolation precautions, while the patients from Niguarda-Milano, INMI-Rome and Padua provided written informed consent.

## Results

Preliminary exploration analyses in the whole dataset of 323 individuals treated with tocilizumab at the Clinics of Infectious Diseases of Modena, Reggio Emilia and Bologna identified the day-4 values for the markers to have the strongest association with the primary outcome (S1 Table). We consequently restricted the analysis to the subset of 266 patients who at day 6 after starting the treatment were still alive and free from mechanical ventilation and for whom day-4 $PaO_2/FiO_2$ ratio, platelets and C-reactive protein (CRP) values were also available. By day 28 of hospital follow-up post treatment initiation, 40 of these (15%) were put under mechanical ventilation or died. Of these events, 26 (10%) were deaths.

Table 1 shows the main demographic characteristics and average markers values recorded at baseline, day 4 and day 9 after starting tocilizumab, as well as day-4 and day-9 marker changes from baseline. The $PaO_2/FiO_2$ ratio was greater in controls at baseline and remained stable over day0-day9 while in controls there was an appreciable deterioration over time from 221 to 157 mmHg (a value approximating the indication for mechanical ventilation) (Table 1, S2 Fig).

In a screening univariable analysis, gender and day-4 $PaO_2/FiO_2$ ratio, platelets and CRP were the factors showing the strongest association with the composite outcome of day-28 mechanical ventilation or death. Baseline SOFA score and respiratory rates, other markers of COVID-19 disease severity, were also strongly associated with the risk of outcome but were not further considered for the construction of the tocilizumab response prediction score because too correlated with the day-4 $PaO_2/FiO_2$ ratio value. Other markers considered (IL-6, D-dimer and total lymphocytes at the various time-points) showed less strong associations and were also discarded at this stage.

Table 2 shows the magnitude of the effect of the four factors selected at univariable analysis from fitting a logistic regression model of the day-28 odds of developing the composite event. In the unadjusted analysis, all factors were associated with at least a 2-fold difference in risk of developing the outcome, results which were confirmed after controlling for gender alone and after full mutual adjustment (Table 2).

When these variables were fitted as binary variables classifying patients according to whether they had a value above or below the median of the study population values, day-4 $PaO_2/FiO_2$ ratio showed the largest association with an aOR of developing the composite end-point of 18.9 (95% CI: 4.14–86.6, p<0.001, Table 3) comparing people with values below and above the median of 209 mmHg in $PaO_2/FiO_2$.

The prediction score was constructed as a linear predictor of the four factors identified in the screening phase with sex as a binary variable and the 3 biomarkers fitted in the log10 scale. The area under the curve (AUC) of the ROC curve using the training dataset was AUC = 0.89, showing a good trade-off between sensitivity and specificity (S1 Fig). As expected, as the performance of the model is overestimated in training, the AUC was significantly better than a random classification of the participants (Mann-Whitney test vs. AUC = 0.5, p-value<0.001). Of note, the internal validation, by means of CV, provided only a slightly lower value for AUC = 0.87 (Fig 1).

On the basis of the estimates of the logistic regression analysis, a simplified prediction score has been developed allocating weights to each of the four components of the score. Thus, a person with a day-4 platelet value above the median was given a score of +3, female gender and a day-4 CRP value above the median both a score of +4 and a day-4 $PaO_2/FiO_2$ ratio value above the median the largest weight of +6 (Table 4). The final score for a person is then obtained by summing the weights of each of the four variables. This sum total score was used to create three risk groups as follows: low risk (total score 0–4, 0–10% risk), intermediate risk (total

**Table 1. Mean of biomarkers by case-control status.**

| Markers | Case-Control status | | | |
| --- | --- | --- | --- | --- |
| | Mech Ventilation-Death | Free of event | p-value* | Total |
| | N = 40 | N = 226 | | N = 266 |
| *Markers, Mean (SD)* | | | | |
| Female, n(%) | 5 (12.5%) | 84 (37.2%) | 0.002 | 89 (33.5%) |
| Age, years | 69 (8) | 63 (13) | 0.005 | 64 (13) |
| SOFA Score | 3 (2) | 2 (1) | < .001 | 2 (1) |
| *PaO$_2$/FiO$_2$ mmHg* | | | | |
| Baseline | 193.6 (113.4) | 241.5 (101.2) | 0.010 | 234.2 (104.3) |
| Day 4 | 124.1 (79.3) | 244.1 (108.0) | < .001 | 225.2 (112.7) |
| Day 9 | 152.9 (116.6) | 247.1 (117.1) | 0.002 | 231.1 (121.8) |
| Change from baseline at Day 4 | -96.3 (106.0) | 18.4 (98.7) | < .001 | 0.0 (108.1) |
| Change from baseline at Day 9 | -77.5 (148.2) | 23.6 (126.7) | 0.005 | 7.1 (135.0) |
| *Respiratory rate* | | | | |
| Baseline | 25.3 (7.2) | 21.7 (5.7) | 0.002 | 22.2 (6.0) |
| Day 4 | 25.4 (5.5) | 21.1 (7.8) | 0.024 | 21.5 (7.7) |
| Day 9 | 23.3 (5.7) | 19.9 (8.8) | 0.182 | 20.3 (8.6) |
| Change from baseline at Day 4 | 2.4 (7.2) | -1.0 (9.4) | 0.159 | -0.6 (9.2) |
| Change from baseline at Day 9 | 0.8 (8.2) | -1.9 (10.0) | 0.365 | -1.5 (9.8) |
| *IL-6, pg/ml* | | | | |
| Baseline | 318.6 (210.4) | 318.1 (430.8) | 0.997 | 318.2 (410.2) |
| Day 4 | 2210 (282.6) | 797.8 (726.2) | < .001 | 910.8 (799.2) |
| Day 9 | 1323 (1382) | 686.6 (755.2) | 0.262 | 713.6 (777.1) |
| Change from baseline at Day 4 | 1783 (399.5) | 460.3 (693.1) | < .001 | 578.4 (770.2) |
| Change from baseline at Day 9 | 1181 (1525) | 217.5 (880.1) | 0.155 | 274.2 (922.5) |
| *D-dimer, mg/dl* | | | | |
| Baseline | 1121 (1647) | 1323 (3346) | 0.819 | 1302 (3213) |
| Day 4 | 3066 (7028) | 2467 (4098) | 0.670 | 2522 (4409) |
| Day 9 | 2649 (3615) | 2423 (4128) | 0.864 | 2453 (4043) |
| Change from baseline at Day 4 | 1485 (8159) | 567.8 (4748) | 0.594 | 658.6 (5134) |
| Change from baseline at Day 9 | 2295 (4710) | 582.5 (6511) | 0.534 | 745.6 (6351) |
| *CRP, mg/dL* | | | | |
| Baseline | 12.9 (7.7) | 10.0 (7.6) | 0.035 | 10.5 (7.7) |
| Day 4 | 5.4 (5.0) | 3.2 (3.8) | 0.005 | 3.6 (4.1) |
| Day 9 | 7.1 (11.5) | 1.4 (3.2) | < .001 | 2.1 (5.5) |
| Change from baseline at Day 4 | -7.3 (8.9) | -6.8 (8.7) | 0.759 | -6.9 (8.7) |
| Change from baseline at Day 9 | -4.6 (14.8) | -8.8 (8.6) | 0.069 | -8.2 (9.7) |
| *Tot Lymphocytes, cells/mm$^3$* | | | | |
| Baseline | 335.2 (616.7) | 612.5 (947.4) | 0.112 | 571.3 (910.1) |
| Day 4 | 642.6 (1321) | 819.2 (1146) | 0.441 | 792.7 (1172) |
| Day 9 | 1025 (1852) | 1110 (1607) | 0.816 | 1097 (1642) |
| Change from baseline at Day 4 | 78.3 (882.6) | 62.3 (728.4) | 0.919 | 64.6 (750.0) |
| Change from baseline at Day 9 | -104 (561.6) | 395.6 (1013) | 0.033 | 324.2 (975.5) |
| *Platelets, cells/mm$^3$* | | | | |
| Baseline | 198.1 (97.7) | 244.5 (111.5) | 0.020 | 237.7 (110.6) |
| Day 4 | 236.9 (113.2) | 337.0 (144.5) | < .001 | 322.3 (144.6) |
| Day 9 | 210.8 (92.7) | 348.0 (141.8) | < .001 | 328.5 (143.9) |
| Change from baseline at Day 4 | 31.4 (82.3) | 92.7 (98.9) | 0.001 | 83.6 (98.9) |

*(Continued)*

**Table 1.** (Continued)

| Markers | Case-Control status | | | |
| --- | --- | --- | --- | --- |
| | Mech Ventilation-Death | Free of event | p-value[*] | Total |
| Change from baseline at Day 9 | 0.4 (104.0) | 119.7 (127.3) | < .001 | 104.0 (130.6) |

[*]Chi2 for gender and unpaired t-test

score 5–9, 10–20% risk) and high risk (total score ≥10, >20% risk). Thus, for a virtual participant who is male (+4), with a CRP> 1.24 mg/dL (+4), a $PaO_2/FiO_2$ ratio >210 mmHg (0) and platelets> 334 cells/mm$^3$ (0) the sum risk score would be 4+4 = +8, placing her in the intermediate risk category. The exact propensity risk for this person estimated from the logistic regression model is 12.4% (it can be calculated from the formula (1) described in the Methods and shown again at the bottom of Table 4).

On average, 40/106 (38%, 95% CI 29%–48%) of individuals in the training dataset classified as having a high risk of mechanical ventilation or death experienced this composite event by day 28 vs. 15% and 4% in the intermediate and low risk groups, respectively.

The predictive value of the proposed score was higher when we used death alone as the endpoint (n = 26 events). AUC was 0.94 in training and 0.92 in cross-validation (S3 Fig).

We also put together an external validation dataset which included patients treated with tocilizumab in 3 independent clinics in Italy and satisfying the same inclusion criteria and endpoint definitions used for the training population. This included 36 patients treated at INMI Spallanzani Roma, 29 at Niguarda hospital Milano and 20 at Padua hospital for a total of 85 additional participants. In this set, 17% were females, median (IQR) $PaO_2/FiO_2$ ratio, platelets and CRP were 223 mmHg (172–329), 294 cells/mm$^3$ (248–383) and 1.0 mg/dL (0.4–4.9), respectively. In total, 11 people experienced mechanical ventilation or death, of whom 7 died. AUC in test were 0.70 for the composite endpoint (Fig 2) and 0.69 for death, showing a larger decrease in accuracy from 0.92.

Finally, we looked at the cumulative probability of developing the composite event starting from 4 days after initiation tocilizumab by means of Kaplan-Meier method. S4 Fig shows that by day 7 post tocilizumab initiation (3 days after baseline) only 22 participants (9%; 95% CI:5%-12%) have already progressed to the composite endpoint. Therefore, in a similar setting,

**Table 2. OR from fitting a logistic regression model.**

| | OR of Mechanical ventilation death | | | | | |
| --- | --- | --- | --- | --- | --- | --- |
| | Unadjusted[*] OR (95% CI) | p-value | Adjusted1[*] OR (95% CI) | p-value | Adjusted2[&] OR (95% CI) | p-value |
| *CRP, per log10 mg/dL higher* | | | | | | |
| Day 4[#] | 3.88 (1.71, 8.78) | 0.001 | 3.66 (1.58, 8.49) | 0.003 | 6.55 (1.93, 22.27) | 0.003 |
| *Gender* | | | | | | |
| F vs. M | 0.24 (0.09, 0.64) | 0.004 | | | 0.33 (0.09, 1.16) | 0.083 |
| *PaO₂/FiO₂ per log10 lower* | | | | | | |
| Day 4[#] | 155.9 (25.42, 956.2) | < .001 | 201.2 (28.49, 1421) | < .001 | 308.8 (30.85, 3091) | < .001 |
| *Platelets, per log10 mg/l lower* | | | | | | |
| Day 4[#] | 23.96 (4.39, 130.7) | < .001 | 21.48 (4.02, 114.7) | < .001 | 15.28 (1.24, 187.7) | 0.033 |

[*]adjusted for gender

[&]adjusted for gender, $PaO_2/FiO_2$ and platelets

[#]time window [+2;+6] days from starting tocilizumab

**Table 3. OR from fitting a logistic regression model.**

| | Case | Control | OR of Mechanical ventilation death | | | | | |
|---|---|---|---|---|---|---|---|---|
| | | | Unadjusted* OR (95% CI) | p-value | Adjusted1* OR (95% CI) | p-value | Adjusted2& OR (95% CI) | p-value |
| | N = 33 | N = 191 | | | | | | |
| **CRP, mg/dL** | | | | | | | | |
| Day 4 | | | | < .001 | | 0.002 | | 0.001 |
| 0–1.23 | 7 (21.2%) | 105 (55.0%) | 1 | | 1 | | 1 | |
| 1.24+ | 26 (78.8%) | 86 (45.0%) | 4.53 (1.88, 10.95) | | 4.18 (1.71, 10.18) | | 6.09 (2.03, 18.28) | |
| **PaO₂/FiO₂ ratio, mmHg** | | | | | | | | |
| Day 4 | | | | < .001 | | < .001 | | < .001 |
| 210+ | 2 (6.7%) | 93 (58.1%) | 1 | | | | 1 | |
| 0–209 | 28 (93.3%) | 67 (41.9%) | 19.43 (4.47, 84.39) | | 17.80 (4.07, 77.76) | | 18.93 (4.14, 86.58) | |
| **Platelets, cells/mm³** | | | | | | | | |
| Day 4 | | | | 0.002 | | 0.004 | | 0.068 |
| 334+ | 8 (24.2%) | 105 (54.4%) | 1 | | | | 1 | |
| 0–333 | 25 (75.8%) | 88 (45.6%) | 3.73 (1.60, 8.68) | | 3.49 (1.48, 8.19) | | 2.56 (0.93, 7.02) | |

*adjusted for gender

&adjusted for gender, PaO₂/FiO₂ ratio and platelets

we are 95% confident that the score could be successfully applied to predict the outcome of a minimum of 88% of the patients who started the drug.

## Discussion

Despite the TESEO study estimates of a substantial reduction in risk of mechanical ventilation in patients treated with SoC+tocilizumab vs. SoC [4], still a non-negligible 15% (40/266) of patients in the tocilizumab group experienced the composite endpoint. The present study gives a tool for early recognition of treatment failure.

We propose a novel predictive model which assumes the knowledge of patients' sex and whether the values of one blood gas analysis and two biomarkers measured at day 4 after tocilizumab administration are above or below a certain threshold: $PaO_2/FiO_2$ ratio <210 mmHg, CRP >1.23 mg/dL and platelets <333 cells/mm³. Our model and risk score had high predictive accuracy and performed similarly in patients treated at different hospitals in Italy, providing independent validation.

As an example, for a virtual male patient who, 4 days after tocilizumab administration, still has relatively preserved respiratory function (e.g. with a $PaO_2/FiO_2$ ratio> 210 mmHg, a value well above the indication for starting invasive mechanical ventilation) the score produces an estimated risk of >20% of failing to respond to the drug if her CRP is >1.24 mg/dL and platelets are >333 cells/mm³ (Table 4).

In support of the composition of our risk score, a recent meta-analysis showed that tocilizumab treatment was associated with reduction in a number of biomarkers including C-reactive protein [18]. Absolute values at baseline did not predict and, although the greatest change was observed at day 9, both absolute values and the changes from baseline at day 9 showed weaker associations with the outcomes and were not included in the algorithm.

Several risk score to predict mortality or respiratory failure in the whole population of patients admitted to hospital with COVID-19 disease have been published, using a large number of variables and a variety of statistical approaches [19–22]. In these studies, detection of clinical worsening after first- or second-line therapies, seemed to be more challenging to

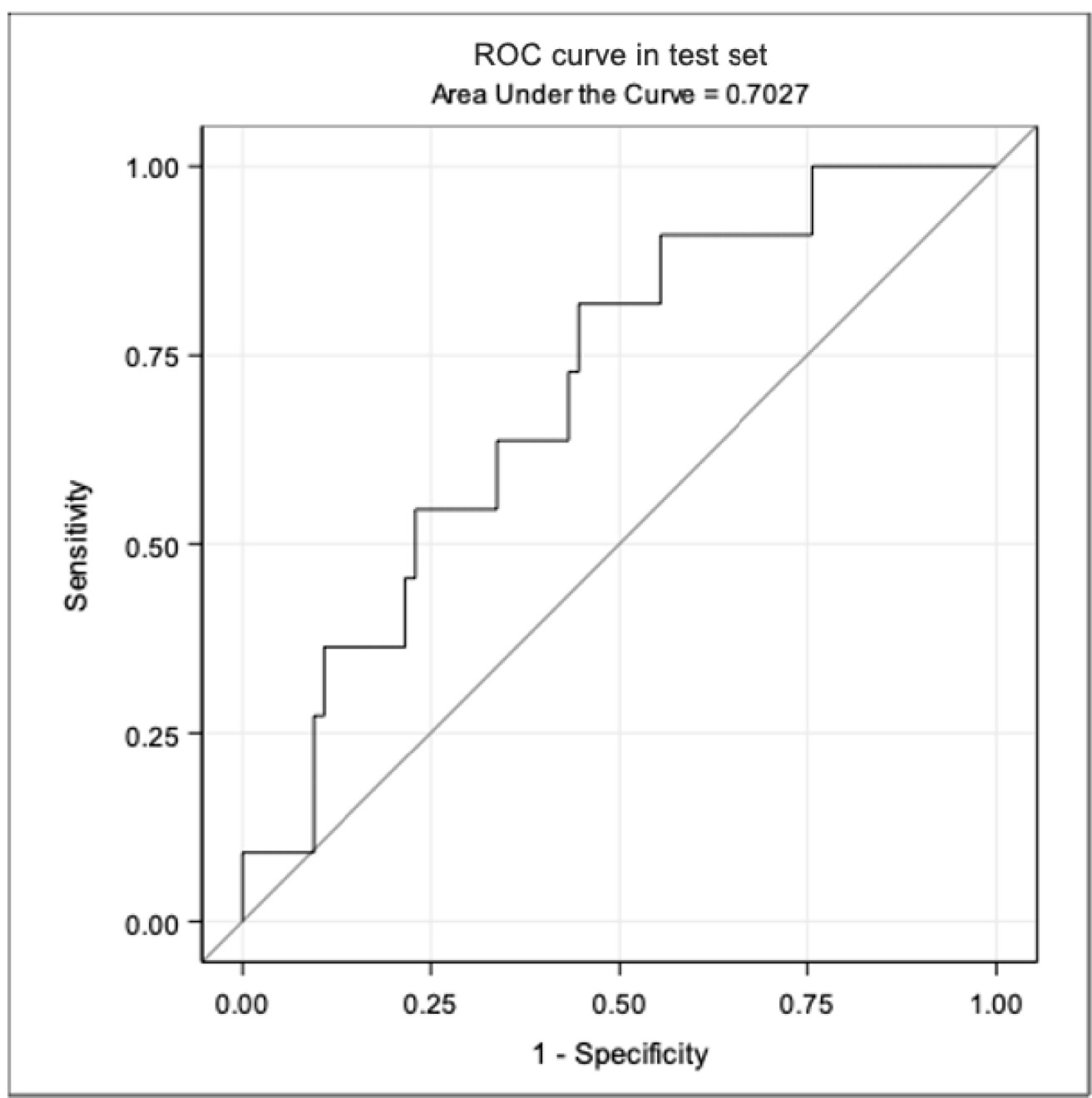

**Fig 1. AUC under the ROC analysis—comparison with CV estimate.**

forecast [19]. To our knowledge, our risk score is the first algorithm derived specifically for people treated with tocilizumab. To build the prediction model, we have used our previous experience in developing risk scores in the field of HIV resistance which was easily transferable to the field of COVID-19 research [23].

**Table 4. Risk score for mechanical ventilation/death on tocilizumab: Exact and simplified risk score.**

**A)**

| Characteristic | Coefficient of logistic regression | Simplified individual score | Example for virtual participant | | | |
|---|---|---|---|---|---|---|
| | | | Observed characteristic | Contribution | Exact propensity score from model | Total Score |
| **Gender** | | | | | | |
| Female | 0 | 0 | | | | |
| Male | 0.91 | +4 | X | +4 | | |
| **CRP mg/dL Day 4** | | | | | | |
| 0–1.24 | 0 | 0 | | | | |
| 1.24+ | 0.99 | +4 | X | +4 | | |
| **PaO$_2$/FiO$_2$ ratio mmHg** | | | | | | |
| 210+ | 0 | 0 | X | 0 | | |
| 0–209 | 1.74 | +6 | | | | |
| **Platelets/mm$^3$** | | | | | | |
| 334+ | 0 | 0 | X | 0 | | |
| 0–333 | 0.71 | +3 | | | | |
| **Total score** | | | | +8 | 12.4% | +8 |

A male with CRP above the median but PaO$_2$/FiO$_2$ ratio and platelets below the median, has an individual score of +8 which corresponds to an estimated propensity to fail tocilizumab of 12.4%

**B) Propensity score relative to simplified score**

| Simplified Score category | Estimated propensity to fail tocilzumab |
|---|---|
| Low (0–4) | 0–10% |
| Intermediate (5–9) | 10–20% |
| High (10+) | 20%+ |

**C)**

The exact formula to calculate the propensity score for a participant i)

PS(i) = Num(i) / Den(i)

Where

Num(i) = exp (-3.85+0.91*male+0.99*CRP+1.74*PaO2/FiO2 ratio +0.71*PLT)

Den(i) = 1 + exp (-3.85+0.91*male+0.99*CRP+1.74*PaO2/FiO2 ratio +0.71*PLT)

In the example of the virtual participant above:

Num(i) = exp (-3.85+0.91+0.99) = 0.0843; Den (i) = 1 + 0.0843; PS = 12.4%

Our study has several limitations. First, we included a selected population of patients who were still alive and with no need for mechanical ventilation 6 days after starting tocilizumab. This selection may have resulted in an underestimation of the overall mortality risk and our proposed score is only applicable to a similar target population. Of note, the specific inclusion criteria hardly affected the process of variables selection. Second, we cannot rule out that a full machine learning approach with a larger number of parameters as well as their interactions could have led to better values for the AUC in validation and test. For example, the score does not account for extent of pre-existing co-morbidities. On the other hand, the simplicity and transparency should make the score more widely applicable in clinical practice. Third, sample size of the test cohort was not large and the AUC in test for the endpoint death was below 70% which is classified as the minimum for a fair predictive value. Also, data collection was not complete for some biomarkers at all centers and we performed a complete case analysis excluding those with missing data. Another limitation is the fact that the score cannot be calculated before day 4 although the average time from admission to ICU appears to be even shorter

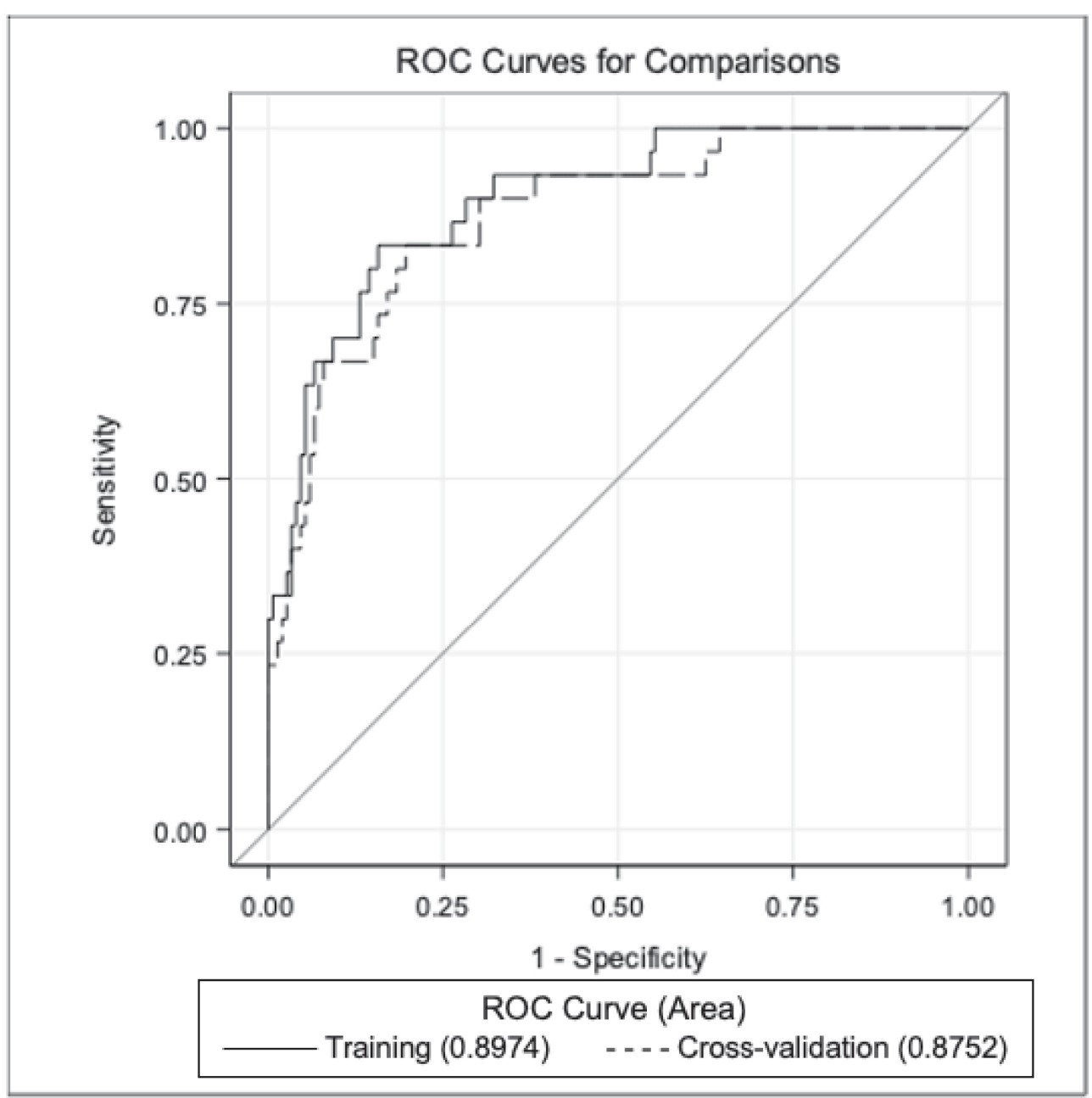

| ROC Model | Area | Standard Error | 95% Wald Confidence Limits | | Somers' D | Gamma | Tau-a |
|---|---|---|---|---|---|---|---|
| Training | 0.8974 | 0.0301 | 0.8384 | 0.9563 | 0.7947 | 0.7947 | 0.2200 |
| Cross-validation | 0.8752 | 0.0343 | 0.8080 | 0.9425 | 0.7504 | 0.7504 | 0.2078 |

**Fig 2. AUC under the ROC for composite endpoint in test set.**

at 3.5 days [16]. Nevertheless, the data from our cohort showed that up to 88% of patients were still free from mechanical ventilation and death by day 4 after starting tocilizumab. However, the calibration performance of the prediction model was not assessed and the score should be further externally validated in the future using other test sets of data collected during the more recent waves of the pandemic. Finally, in the real-life setting people, future target populations are likely to be people who have failed glucocorticoids and not the drugs considered as standard of care at the time of this study (e.g. chloroquine) and this might or might not have an impact on the predictive ability of the score.

In conclusion, our study provides an important tool that could assist clinicians in identifying early in the course of treatment with tocilizumab, patients who are likely to progress to mechanical ventilation or death. This algorithm may also rapidly identify patients who qualify for rescue therapies with new promising strategies such as monoclonal antibodies. Of note, the score highlights the fact that it is not only respiratory function but the combination of different factors routinely tested in clinical practice that could alert clinicians to change strategy. We believe that our proposed score could be easily implemented in clinical practice and has the potential to be useful for many more patients now that tocilizumab is recommended for use in those critically ill.

## Supporting information

**S1 Table. Mean of biomarkers by case-control status in the unselected population.**
(DOCX)

**S1 Fig. AUC under the ROC analysis–training set.**
(DOCX)

**S2 Fig. Average trend in PaO2/FiO2 ratio and biomarkers over day0-day9 by case-control status.**
(DOCX)

**S3 Fig. AUC under the ROC for death in the training set and CV analysis.**
(DOCX)

**S4 Fig. KM estimate of experiencing the composite event past day-4 window.**
(DOCX)

## Author Contributions

**Conceptualization:** Cristina Mussini, Alessandro Cozzi-Lepri, Marianna Meschiari, Erica Franceschini, Giovanni Guaraldi.

**Data curation:** Cristina Mussini, Alessandro Cozzi-Lepri, Marianna Menozzi, Erica Franceschini, Antonello Pietrangelo, Massimo Girardis, Andrea Cossarizza, Roberto Tonelli, Enrico Clini, Marco Massari, Michele Bartoletti, Anna Ferrari, Anna Maria Cattelan, Paola Zuccalà, Miriam Lichtner, Roberto Rossotti, Enrico Girardi, Emanuele Nicastri, Massimo Puoti, Andrea Antinori, Pierluigi Viale, Giovanni Guaraldi.

**Formal analysis:** Alessandro Cozzi-Lepri, Marianna Menozzi.

**Investigation:** Cristina Mussini, Alessandro Cozzi-Lepri, Marianna Menozzi, Marianna Meschiari, Erica Franceschini, Jovana Milic, Lucio Brugioni, Antonello Pietrangelo, Massimo Girardis, Andrea Cossarizza, Roberto Tonelli, Enrico Clini, Michele Bartoletti, Anna

Ferrari, Anna Maria Cattelan, Miriam Lichtner, Roberto Rossotti, Enrico Girardi, Emanuele Nicastri, Massimo Puoti, Andrea Antinori, Pierluigi Viale, Giovanni Guaraldi.

**Methodology:** Cristina Mussini, Alessandro Cozzi-Lepri, Marianna Menozzi, Marianna Meschiari, Marco Massari, Paola Zuccalà, Giovanni Guaraldi.

**Resources:** Cristina Mussini.

**Supervision:** Cristina Mussini, Alessandro Cozzi-Lepri, Lucio Brugioni, Andrea Cossarizza, Enrico Clini, Marco Massari, Anna Maria Cattelan, Miriam Lichtner, Roberto Rossotti, Enrico Girardi, Emanuele Nicastri, Massimo Puoti, Andrea Antinori, Pierluigi Viale, Giovanni Guaraldi.

**Validation:** Cristina Mussini, Alessandro Cozzi-Lepri, Giovanni Guaraldi.

**Visualization:** Cristina Mussini, Alessandro Cozzi-Lepri, Giovanni Guaraldi.

**Writing – original draft:** Cristina Mussini, Alessandro Cozzi-Lepri, Marianna Menozzi, Marianna Meschiari, Jovana Milic, Giovanni Guaraldi.

**Writing – review & editing:** Cristina Mussini, Alessandro Cozzi-Lepri, Marianna Meschiari, Jovana Milic, Anna Maria Cattelan, Miriam Lichtner, Giovanni Guaraldi.

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
