## [Decision Letter · Decision Letter 0]

26 Jan 2021

PONE-D-20-35873

Development and validation of a prediction model for tocilizumab failure in hospitalized patients with SARS-CoV-2 infection

PLOS ONE

Dear Dr. Mussini,

Thank you for submitting your manuscript to PLOS ONE. After careful consideration, we feel that it has merit but does not fully meet PLOS ONE’s publication criteria as it currently stands. Therefore, we invite you to submit a revised version of the manuscript that addresses the points raised during the review process.

We look forward to receiving your revised manuscript.

Kind regards,

Aleksandar R. Zivkovic

Academic Editor

PLOS ONE

4. Please ensure that you refer to Figure 2 in your text as, if accepted, production will need this reference to link the reader to the figure.

Reviewers' comments:

Reviewer's Responses to Questions

**Comments to the Author**

1. Is the manuscript technically sound, and do the data support the conclusions?

Reviewer #1: Yes

Reviewer #2: Yes

Reviewer #3: Partly

2. Has the statistical analysis been performed appropriately and rigorously? 

Reviewer #1: Yes

Reviewer #2: Yes

Reviewer #3: No

3. Have the authors made all data underlying the findings in their manuscript fully available?

Reviewer #1: Yes

Reviewer #2: Yes

Reviewer #3: No

4. Is the manuscript presented in an intelligible fashion and written in standard English?

Reviewer #1: Yes

Reviewer #2: Yes

Reviewer #3: No

5. Review Comments to the Author

Reviewer #1: Dear Authors,

This study was interesting and brought further knowledge in COVID 19 especially in severe disease. My comments are mentioned below

1. Why do you choose female gender as the risk factor of severe. As we knew from many published trial men were strongly risk factors of severity in COVID 19

2.Previous study discussed that Toci could improve biomarker of severe disease. (DOI: 10.1002/jmv.26698 ) This topic should be mentioned in the discussion. Although this study had novelty the first, you could discuss with the comparison of other virus disease or outbreak who develop risk scoring for treatment.

3. Although this scoring probability was internally statistically significant, this scoring system should be testing in external validation with higher and different population, and this should be mentioned in discussion or conclusion.

4. Until now, we have still not known, what is the best time to give toci, may be this one of the factors, why the results of the study were conflicting, and previous study was using Chloroquine which data from Recovery showed harmful. This population also still used Chloroquine, this issue should be considered.

Reviewer #2: The Authors here presented a well conducted and clear study. The statistical method is clear exposed and nicelt designed. All the data are clear. The study limitations are well described in the discussion. Only few comments:

- Line 41: there is another meta-analysis you should cite to reinforce this point: Mortality in tocilizumab-treated patients with COVID-19: a systematic review and meta-analysis. Clin Exp Rheumatol. 2020 Nov-Dec;38(6):1247-1254. Epub 2020 Dec 3. PMID: 33275094.

- Line 44: to date this affermation is not true, there is at least one RCT you should cite: Tocilizumab Trial Investigators. Efficacy of Tocilizumab in Patients Hospitalized with Covid-19. N Engl J Med. 2020 Dec 10;383(24):2333-2344. doi: 10.1056/NEJMoa2028836. Epub 2020 Oct 21. PMID: 33085857; PMCID: PMC7646626.

- Correct some typo errors

Reviewer #3: 1) To properly evaluate the performance of a prediction model, only using an independent testing dataset is meaningful. It’s great there is an independent dataset available in the study. The matter is then to describe the performance of the model properly and objectively—build the model using the training dataset and evaluate its performance using the testing dataset. Please remove the results of cross-validation on the same dataset to avoid the unnecessarily exaggeration of the model performance.

6. PLOS authors have the option to publish the peer review history of their article (what does this mean?). If published, this will include your full peer review and any attached files.

Reviewer #1: **Yes: **Andree Kurniawan

Reviewer #2: No

Reviewer #3: No

---

## [Author Response · Author response to Decision Letter 0]

1 Feb 2021

Modena 30/01/2021

To:

Academic Editor

PLOS ONE

Dear Editor, 

We are very grateful for forwarding to us the reviewers’ comments to our paper PONE-D-20-35873 entitled: Development and validation of a prediction model for tocilizumab failure in hospitalized patients with SARS-CoV-2 infection

We here provide a point-by-point reply to the major and minor comments and we have incorporated the related changes in the manuscript. 

Editor

We checked the PLOS ONE style templates and we believe this revised version meets the journal requirements. 

We have now included a supplementary Table showing the results of the preliminary screening of the association between the markers value at baseline, day4 and day9 with the outcome in the whole dataset of 323 participants (Supplementary Table 1). 

Professor Cristina Mussini is the corresponding author who has a ORCID iD that is validated in Editorial Manager.

 4. Please ensure that you refer to Figure 2 in your text as, if accepted, production will need this reference to link the reader to the figure.

We apologize for the omission. Old Figure 2 is now Figure 1 and a reference has been added at line 205 of the revised version.

Reviewers' comments:

Comments to the Author

Reviewer #1: 

Dear Authors,

This study was interesting and brought further knowledge in COVID 19 especially in severe disease. My comments are mentioned below

1. Why do you choose female gender as the risk factor of severe. As we knew from many published trial men were strongly risk factors of severity in COVID 19

We included sex in the score. The reviewer is absolutely right that males are at a higher risk than female. There was indeed a typo at lines 214 (in the revised version line 276) and 252 (in the revised version line 322) in the calculation of the risk score for a virtual patient and a score of +4 should be given to someone of male sex, not female (this is shown correctly in Table 4). 

2. Previous study discussed that Toci could improve biomarker of severe disease. (DOI: 10.1002/jmv.26698) This topic should be mentioned in the discussion. Although this study had novelty the first, you could discuss with the comparison of other virus disease or outbreak who develop risk scoring for treatment.

We have now added the suggested reference and the following sentence has been incorporated in the Discussion section (lines 279-281):

“In support of the composition of our risk score, a recent meta-analysis showed that tocilizumab treatment was associated with reduction in a number of biomarkers including C-reactive protein (18).”

We indeed had built up our expertise in developing risk scores in the field of HIV resistance and we have also mentioned this in the Discussion (lines 337-339):

“To build the prediction model, we have used our previous experience in developing risk scores in the field of HIV resistance which was easily transferable to the field of COVID-19 research (23).”

3. Although this scoring probability was internally statistically significant, this scoring system should be testing in external validation with higher and different population, and this should be mentioned in discussion or conclusion.

We did validate the performance of the risk score on an external dataset of data collected at clinics not included in the training set although data were shown as supplemental material. We have now removed the performance of the model in training and replaced those results with a Figure showing the performance of the model in the external test set. We agree with the reviewer that the sample size of the test is small and we should aim to validate the score in the future on other larger sets. We have added a sentence in the Discussion regarding this future work (lines 381-382):

“However, the score should be further externally validated in the future using other test sets of data collected during the more recent waves of the pandemic.”

4. Until now, we have still not known, what is the best time to give toci, may be this one of the factors, why the results of the study were conflicting, and previous study was using Chloroquine which data from Recovery showed harmful. This population also still used Chloroquine, this issue should be considered.

Indeed, tocilizumab was added to chloroquine in most participants as it was standard of care at the time but it was an early use and the drug was discontinued promptly as it did not show much benefit. Unfortunately, as few patients were not using chloroquine, the effect of tocilizumab affected by use of this drug could not be tested in our study. We have now specifically mentioned chloroquine in line 385 of the revised version. We agree that trials addressing the question ‘when is best to treat with tocilizumab’ are lacking. However, recent evidence from trials have led to a change of treatment guidelines in the UK, that are now recommending the use of tocilizumab in critical patients. We have planned a trial at University of Modena and Reggio Emilia to specifically address this question.

 Reviewer #2: 

The Authors here presented a well conducted and clear study. The statistical method is clear exposed and nicelt designed. All the data are clear. The study limitations are well described in the discussion. Only few comments:

- Line 41: there is another meta-analysis you should cite to reinforce this point: Mortality in tocilizumab-treated patients with COVID-19: a systematic review and meta-analysis. Clin Exp Rheumatol. 2020 Nov-Dec;38(6):1247-1254. Epub 2020 Dec 3. PMID: 33275094.

The literature on tocilizumab indeed grows very quickly. We thank you the reviewer for pointing out the new meta-analysis. We have revised the state of art on tocilizumab treatment and added some more recent published meta-analyses and randomized trials. The following sentences have been added or modified in the Introduction:

“These results were confirmed by 2 more recently published meta-analyses including a larger number of studies (8,9).” – lines 87-88

“A pharmaceutical company press releases showed no evidence for clinical improvement and reduced risk of mortality when comparing patients treated with tocilizumab vs. placebo in the double-blind COVACTA trial, but final results remain unpublished (10). Similar negative results were obtained prescribing tocilizumab to patients with mild to moderate COVID19 pneumonia in the attempt to prevent the cytokine storm (11, 12).” – lines 90-95

“More recently, the REMAP_CAP trial conducted in critically ill patients showed a reduction in mortality in patients receiving either tocilizumab or sarilumab. As a consequence of these results, tocilizumab is now recommended for use in the clinics for patients in critical conditions in the UK (15).” – lines 102-114

- Line 44: to date this affermation is not true, there is at least one RCT you should cite: Tocilizumab Trial Investigators. Efficacy of Tocilizumab in Patients Hospitalized with Covid-19. N Engl J Med. 2020 Dec 10;383(24):2333-2344. doi: 10.1056/NEJMoa2028836. Epub 2020 Oct 21. PMID: 33085857; PMCID: PMC7646626.

See our response to previous point. At the time of submission (late November 2020), none of the ongoing trials on tocilizumab had been published. We thank you the reviewer for point out the trial published on NEJM. We have revised sentences concerning the state of art on tocilizumab treatment and added the reference for this and other recent published trials.

- Correct some typo errors

We have carefully checked the whole text for typos and corrected them.

Reviewer #3: 

To properly evaluate the performance of a prediction model, only using an independent testing dataset is meaningful. It’s great there is an independent dataset available in the study. The matter is then to describe the performance of the model properly and objectively—build the model using the training dataset and evaluate its performance using the testing dataset. Please remove the results of cross-validation on the same dataset to avoid the unnecessarily exaggeration of the model performance.

We agree with the reviewer that the performance on the test set is key. We have therefore removed Figure 1 of the performance on training (which we agree is misleading) and replaced with Supplementary Figure 3 with the performance on test. Regarding the data obtained in cross-validation, we respectfully disagree with the reviewer. Cross-validation is a good estimator of the extra-sample error distribution and especially in our case, in which the sample size of the test set is small, it is a crucial step and represents even a more robust validation if no population shift or selection bias is expected. Indeed, the TRIPOD statement for model predictions indicate Internal validation as a necessary part of model development [Steyerberg EW: Clinical Prediction Models: A Practical Approach to Development, Validation, and Updating. New York: Springer; 2009.; Collins GS, Reitsma JB, Altman DG, Moons KG. Transparent reporting of a multivariable prediction model for individual prognosis or diagnosis (TRIPOD): the TRIPOD statement. BMJ. 2015 Jan 7;350:g7594.].

We hope that this revised version of the paper is now suitable for publication in PLOS ONE and we look forward to hearing back from you.

Best regards,

Prof. Cristina Mussini

---

## [Editor Report · Decision Letter 1]

4 Feb 2021

Development and validation of a prediction model for tocilizumab failure in hospitalized patients with SARS-CoV-2 infection

PONE-D-20-35873R1

Dear Dr. Mussini,

We’re pleased to inform you that your manuscript has been judged scientifically suitable for publication and will be formally accepted for publication once it meets all outstanding technical requirements.

Kind regards,

Aleksandar R. Zivkovic

Academic Editor

PLOS ONE

---

## [Editor Report · Acceptance letter]

11 Feb 2021

PONE-D-20-35873R1 

Development and validation of a prediction model for tocilizumab failure in hospitalized patients with SARS-CoV-2 infection 

Dear Dr. Mussini:

I'm pleased to inform you that your manuscript has been deemed suitable for publication in PLOS ONE. Congratulations! Your manuscript is now with our production department. 

Kind regards, 

on behalf of

Dr. Aleksandar R. Zivkovic 

Academic Editor

PLOS ONE